# Opioid Use in Cancer Pain Management: Navigating the Line Between Relief and Addiction

**DOI:** 10.3390/ijms26157459

**Published:** 2025-08-01

**Authors:** Maite Trullols, Vicenç Ruiz de Porras

**Affiliations:** GRET and Toxicology Unit, Department of Pharmacology, Toxicology and Therapeutic Chemistry, Faculty of Pharmacy and Food Sciences, University of Barcelona, 08028 Barcelona, Spain; maitetrullolsmo02@gmail.com

**Keywords:** opioids, cancer-related pain, opioid use disorder, pharmacogenetics, pain management, personalized medicine, addiction prevention

## Abstract

The use of opioids for cancer-related pain is essential but poses significant challenges due to the risk of misuse and the development of opioid use disorder (OUD). This review takes a multidisciplinary perspective based on the current scientific literature to analyze the pharmacological mechanisms, classification, and therapeutic roles of opioids in oncology. Key risk factors for opioid misuse—including psychiatric comorbidities, prior substance use, and insufficient clinical monitoring—are discussed in conjunction with validated tools for pain assessment and international guidelines. The review emphasizes the importance of integrating toxicological, pharmacological, physiological, and public health perspectives to promote rational opioid use. Pharmacogenetic variability is explored as a determinant of treatment response and addiction risk, underscoring the value of personalized medicine. Evidence-based strategies such as early screening, psychosocial interventions, and the use of buprenorphine-naloxone are presented as effective measures for managing OUD in cancer patients. Ultimately, this work advocates for safe, patient-centered opioid prescribing practices that ensure effective pain relief without compromising safety or quality of life.

## 1. Introduction

### 1.1. Cancer: An Overview

Cancer is a complex disease characterized by abnormal and uncontrolled cell growth resulting from the loss of regulatory mechanisms governing cell proliferation, differentiation, and programmed cell death (apoptosis). These cells can proliferate indefinitely, invade surrounding tissues, migrate to distant parts of the body, and induce the formation of new blood vessels (angiogenesis) to secure the nutrients and oxygen required to sustain their growth. Moreover, they often exhibit defects in DNA repair mechanisms and can evade the host immune response, which further contributes to their survival and expansion [1,2].

As these cells continue to divide, they form an abnormal mass of tissue known as a tumor. However, the term “tumor” generally refers to any abnormal tissue growth, which may be either benign or malignant in nature [1,2,3]. When tumor cells acquire the ability to disseminate through the lymphatic or circulatory systems and colonize distant organs or tissues, the process is known as metastasis—one of the main causes of cancer-related mortality [1,2,4].

In 2022, the International Agency for Research on Cancer (IARC) published estimates that enable analysis of cancer incidence on a global scale [5]. Cancer has been identified as one of the most serious challenges to both public health and the global economy in the 21st century, accounting for one in six deaths (16.85%) and one in four deaths attributed to non-communicable diseases (22.8%). Moreover, it is the third leading cause of death among 30- to 69-year-olds in 177 of the 183 countries analyzed [6]. In addition to being a major barrier to increasing life expectancy, cancer has a substantial economic impact, with costs that vary depending on the type of cancer, geographical location, and gender [7].

The ten most prevalent cancer types across both sexes account for 60% of new diagnoses and cancer-related deaths worldwide. Lung cancer is the most frequently diagnosed (12% of all cases), followed by breast cancer (12%), colorectal cancer (10%), prostate cancer (7%), and stomach cancer (5%) [5]. In terms of mortality, lung cancer also ranks first worldwide (19% of cancer-related deaths), followed by colorectal cancer (9%), liver cancer (8%), breast cancer (7%), and stomach cancer (7%) [5]. On the other hand, regarding sex-specific incidence, breast cancer has the highest incidence among women and is also the leading cause of cancer-related death, followed by colorectal and lung cancers. In contrast, among men, lung cancer shows the highest incidence and mortality, followed by prostate and colorectal cancers [5]. According to data from the GLOBOCAN 2022 study, approximately one in five men and one in five women will develop cancer during their lifetime, while one in nine men and one in twelve women will die from the disease [5].

### 1.2. Fundamentals of Pain

According to the International Association for the Study of Pain (IASP), pain is defined as an unpleasant sensory and emotional experience associated with actual or potential tissue damage. However, patients may experience pain even in the absence of observable tissue damage. For this reason, pain is considered a subjective experience influenced by multiple physical, psychological, and social factors, which makes its assessment particularly challenging [8,9].

Once the nature of pain has been defined, it is essential to understand the different ways in which it can manifest. The classification of pain, based on its temporal evolution and physiopathological origin, allows for a better understanding of the underlying mechanisms and facilitates the implementation of appropriate therapeutic strategies [8,9,10].

According to its temporal evolution [8,9,10]:

**Physiological pain** occurs in the absence of tissue damage and results from brief stimulation of nociceptors in the skin or other tissues. It is considered a protective sensation essential for patient survival and well-being.

**Acute pain** is characterized by the presence of tissue damage, and its duration depends on the healing time of the affected tissues. It is important to note that psychological factors can influence its progression and contribute to the development of chronic pain.

**Chronic pain** results from the persistent stimulation of nociceptors in previously injured tissues.

According to its origin [8,9,10]:

**Somatic, inflammatory, or nociceptive pain** arises from the activation of nociceptors in response to tissue damage. It can occur in any part of the body, except for the central nervous system (CNS).

**Neuropathic pain** is characterized by lesions or dysfunctions of the peripheral or central nervous system. Neoplasms are a common cause of this type of pain due to compression exerted by abnormal tissue growth on nearby nerves.

### 1.3. Cancer-Related Pain

Cancer-related pain (CRP) refers to pain experienced by individuals diagnosed with a malignant tumor, arising as a direct or indirect consequence of the disease. As such, CRP is considered a symptom associated with cancer and, in certain cases, may represent the first sign of the disease, prompting the patient to seek medical attention and potentially leading to the tumor’s diagnosis [10,11,12,13].

The intensity of CRP can range from moderate to severe, and both its frequency and severity are often closely linked to disease progression [10,11,12,13]. This type of pain may have diverse etiologies, characteristics, and physiopathological mechanisms, and it is not always caused solely by the tumor itself. In many cases, multiple contributing factors may coexist, such as local tumor growth with nerve compression, bone metastases, adverse effects of oncological treatments (including chemotherapy, radiotherapy, or surgery), as well as associated infectious or inflammatory processes [10,11,12,13] (Figure 1).

Accordingly, based on its origin, the etiology of CRP can be classified into three main categories: pain directly or indirectly caused by the cancer itself, which accounts for approximately 85% of cases; pain resulting from antineoplastic treatment, representing about 17% of cases; and pain caused by other concomitant disorders unrelated to cancer, comprising around 10% of cases [10,11,12,13] (Figure 1).

Pain is one of the most disabling symptoms experienced by cancer patients. According to available data, it affects approximately 40% of patients after treatment, 55% during treatment, and up to 66% of patients in advanced, metastatic, or terminal stages of the disease [10,11,12,13] (Figure 1).

The assessment of CRP is essential to characterize the type of pain and identify potential underlying physiopathological mechanisms, with the aim of guiding the most appropriate therapeutic decisions. A detailed understanding of various pain characteristics—such as intensity, duration, severity, and temporal variations—enables better individualization of treatment [11,14]. Therefore, both patient-specific and pain-related characteristics are key elements in selecting therapeutic strategies and are often used to classify pain in different clinical settings [11,15].

Pain intensity is one of the most relevant parameters for the classification and evaluation of CRP. Healthcare professionals often rely on this variable to determine the appropriate treatment approach [11]. To assess pain intensity, several methods are available, with the Numeric Rating Scale (NRS) being one of the most widely used. This scale categorizes pain severity as follows: NRS 1–4 indicates mild pain, NRS 5–6 indicates moderate pain, and NRS 7–10 indicates severe pain [11] (Figure 1).

Another way to classify CRP is based on its anatomical location, as cancer can affect virtually any body tissue. The most common sites where pain manifests include soft tissues (45% of cases), bones (35%), nerves (34%), and viscera (33%) (11, 12). It is important to note that, in cases of metastatic cancer, pain often extends across multiple anatomical regions [11,12] (Figure 1).

Finally, it is also crucial to consider individual patient characteristics such as age, cognitive function, and psychological factors. Thus, the assessment of pain intensity should form part of a comprehensive clinical evaluation to inform appropriate therapeutic decisions (Figure 1).

### 1.4. Overview of Opioids

Opioids are the mainstay and most effective analgesic treatment for managing acute, perioperative, and chronic pain. Cancer patients often require opioids to control pain arising both from the disease itself and from the treatments they undergo [16]. However, it has been observed that opioid use can lead to dependence and potential misuse. This phenomenon has led to stricter regulations on opioid prescriptions in recent years [17]. In this context, inappropriate use and abuse of opioids have resulted in a public health crisis in several developed countries, with an increase in emergency department visits and a greater burden on healthcare systems [17].

Opioids are classified into endogenous opioids, produced naturally by the body, and exogenous opioids, which include drugs and substances that mimic their action. Both types act on the same receptors in the nervous system and play a key role in pain modulation [16].

#### 1.4.1. The Endogenous Opioid System

Endogenous opioids play a key role in regulating processes such as emotions, memory, and analgesia. They are also involved in modulating various biological functions, including embryonic development, blood vessel formation (angiogenesis), intestinal peristalsis, and respiratory control, among others [18,19,20,21].

The relevance of opioid receptors in the brain’s reward and addiction circuitry is particularly noteworthy. These receptors serve as a common substrate for most drugs of abuse and play a central role in reinforcing physical dependence. Opioids inhibit GABAergic neurons in the ventral tegmental area (VTA), leading to dopamine release in the nucleus accumbens. This mechanism is fundamental to the addictive process induced by drugs [18,19,20,21].

As a result, several risk factors—such as alcohol or tobacco use, depression, anxiety, use of other illicit drugs, or adverse childhood experiences—have been associated with an increased susceptibility to prescription opioid misuse during cancer treatment [22].

##### Endogenous Opioid Receptors

The endogenous opioid system consists of four G protein-coupled receptors (GPCRs), each containing seven transmembrane domains. These receptors are μ (mu) receptors, encoded by the *OPRM1* gene; δ (delta) receptors, encoded by the *OPRD1* gene; κ (kappa) receptors, encoded by the *OPRK1* gene; and nociceptin receptors, encoded by the *OPRL1* gene [16,23].

Although each receptor is encoded by a distinct gene, they share more than 60% similarity in their amino acid composition [16,23]. Notably, each receptor can bind to multiple endogenous ligands, some of which produce similar effects, while others elicit distinct responses [18,21,23].

In this context, several studies have investigated the role of different opioid receptors in mediating the effects of certain drugs and medications, such as morphine. To this end, genetically modified mice (knockout models) lacking one or more of these receptors have been used. This approach allows for the observation of the effects resulting from the absence of a specific receptor. Specifically, various tests have been conducted to assess analgesic efficacy and addiction-related phenomena, such as self-administration, place preference, and withdrawal syndrome. However, it is important to consider that the organism may compensate for the absence of these receptors through alternative adaptive mechanisms [24,25].

In this regard, it has been observed that, in the absence of the µ receptor, no analgesic effect nor any of the phenomena associated with addiction occur. This indicates that the µ receptor is essential for both the analgesic and addictive effects of morphine [26].

Regarding the δ receptor, the analgesic effect is maintained, and withdrawal syndrome is also observed. However, studies based on self-administration and place preference have not provided conclusive results. These observations suggest that the δ receptor is not essential for the analgesic effect but could be involved in the mechanisms of physical dependence [20].

In the case of the κ receptor, studies show that animals continue to experience an analgesic effect, display place preference, and present withdrawal symptoms, although with reduced intensity. This suggests that the κ receptor is not required to produce analgesic effects but plays a modulatory role in the severity of withdrawal [27].

It is important to highlight that the expression pattern of opioid receptors in the nervous system varies depending on the receptor type. In this regard, structural studies using crystallography have been carried out to analyze the molecular differences among the various receptors. These studies have enabled comparisons between inactive conformations and active forms of the receptors, with the aim of identifying potential binding sites for the design of new opioid analgesics [16].

In conclusion, this evidence provides relevant information on how different agonists alter receptor conformations, modulating pain through intracellular signaling cascades. These findings open the door to the development of more selective and effective pharmacological treatments [16].

##### Endogenous Opioid Ligands

Opioids can be endogenous or exogenous, and both types bind specifically to opioid receptors located in the brain, the spinal cord, and various peripheral tissues, such as the joints and the myenteric plexuses of the digestive system [21,28].

Currently, endogenous opioids are classified into four main families: enkephalins, dynorphins, endorphins, and nociceptin/orphanin FQ [28]. These peptides originate from protein precursors (in the form of prepro- and pro-forms), which are converted into their active forms through intracellular maturation processes [16,28].

All of these peptides share a characteristic N-terminal sequence, Tyr-Gly-Gly-Phe-(Met/Leu), which explains how a single precursor can give rise to multiple opioids [16,28]. In this sequence, the Tyr and Phe residues are essential for binding to the opioid receptor, while the two Gly residues act as structural spacers [28].

Moreover, these peptides can undergo post-translational modifications such as acetylation, methylation, or phosphorylation, which may alter their biological activity as well as their affinity and selectivity for receptors, leading to significant changes in their physiological actions [28,29].

#### 1.4.2. Mechanism of Action of Opioids

Opioid receptors, being coupled to G(i/o)-type G proteins, modulate inhibitory cellular responses. Upon activation, they inhibit adenylyl cyclase, leading to decreased levels of cyclic adenosine monophosphate (cAMP) and reduced protein kinase A (PKA) activity, which results in less phosphorylation and, consequently, lower activation of various intracellular proteins, such as voltage-dependent calcium channels. These channels close, thereby decreasing the release of excitatory neurotransmitters like glutamate or substance P at presynaptic nerve terminals. Likewise, at the postsynaptic level, opioids promote the opening of potassium channels, causing membrane hyperpolarization and reducing the likelihood of action potential generation [18,21] (Figure 2).

#### 1.4.3. Classification of Opioid Drugs

Exogenous opioids can be classified according to various criteria, such as their origin, chemical structure, or their affinity and efficacy at opioid receptors. However, the most commonly used classification is based on pharmacological affinity/efficacy, which divides these compounds into the following groups [18,20,21].

**Pure agonists**: These have high affinity for opioid receptors and strong intrinsic activity. Examples in this group include morphine, heroin, pethidine, methadone, fentanyl, and its derivatives.

**Partial agonists**: Their efficacy is lower than that of pure agonists. They only produce analgesic effects when administered alone; in the presence of a pure agonist, they act as antagonists. Buprenorphine is a classic example of this group.

**Mixed agonist–antagonists**: These have limited analgesic effects, acting as agonists at κ receptors and as partial agonists or antagonists at μ receptors. They may trigger withdrawal syndrome in patients treated with pure agonists. Pentazocine, butorphanol, and nalorphine are examples of this type of drug.

**Antagonists**: These bind with high affinity to opioid receptors but do not activate them, thereby blocking the effects of agonists. As they have no intrinsic activity, they do not produce analgesic effects. They are mainly used as antidotes in cases of poisoning or overdose. Naltrexone and naloxone belong to this group.

It is important to highlight that although the terms “opiate” and “opioid” are often used as synonyms, they have different meanings and should be distinguished [20]:

An opiate is any substance obtained from opium, which is extracted from the capsule of the *Papaver somniferum* plant. In contrast, an opioid refers to any compound—natural, synthetic, or semi-synthetic—that has the ability to specifically bind to opioid receptors in the nervous system. For example, morphine is both an opiate (because it is derived from opium) and an opioid (due to its pharmacological action on opioid receptors).

### 1.5. Pathophysiological Mechanism of Opioid Addiction

Addiction is a complex brain disorder related to behavior and is considered chronic, relapsing, and multifactorial. It is defined as a disease characterized by the compulsive seeking and consumption of substances despite negative consequences, caused by persistent alterations in brain circuits involved in motivation, behavioral control, reinforcement, and learning [30].

Opioid addiction is considered a chronic alteration of brain structure and function [31,32] and is associated with a dysregulation of the reward system [32,33,34]. The reward center is located within the dopaminergic pathways of the mesocorticolimbic system (MCLS) [32,35], which originate in the ventral tegmental area (VTA) and project to the nucleus accumbens, the amygdala, and the prefrontal cortex. The main function of this system is to identify, learn about, and respond to rewarding stimuli, which are essential for survival [32,36].

Opioids activate the MCLS by binding to µ-opioid receptors, inhibiting GABAergic neurons, thereby reducing GABA release and leading to a massive and sustained dopamine release in the nucleus accumbens as a consequence of the disinhibition of dopaminergic neurons [24,32]. This excess of dopamine generates sensations of euphoria and well-being, promoting positive reinforcement [24,32] (Figure 3).

However, over time, the brain develops a neuroadaptive response opposite to the euphoric effect [32,37], through activation of the hypothalamic–pituitary–adrenal axis and the cerebral amygdala. This activation leads to increased production of stress-related hormones and peptides [32,38], notably corticotropin-releasing factor [32,39] and dynorphin [32] (Figure 3).

This neurobiological adaptation is responsible for the negative symptoms during withdrawal, such as dysphoria, anxiety, or irritability, which act as negative reinforcement and promote continued use to relieve discomfort [24,32]. Thus, addiction is not only initiated by the pursuit of pleasure, but is maintained and becomes chronic as a mechanism to avoid the negative consequences of withdrawal (Figure 3).

Over time, the pleasurable effect of opioids diminishes, while the compulsive need for consumption increases, consolidating the addictive disorder and raising the risk of relapse, even after prolonged periods of abstinence [32].

## 2. Comprehensive Care and Continuous Management of the Oncology Patient

Oncology is a medical specialty that requires careful planning and constant coordination among various professionals—such as psychologists, surgeons, nurses, and palliative care teams—to ensure maximum treatment effectiveness [40]. The oncologist, due to their specialized knowledge, assumes the leadership of this collaborative process [40].

Comprehensive care is essential throughout the entire course of the disease. Effective symptom control from the early stages can help prevent worsening, while in advanced or terminal stages, it becomes crucial to ensure the patient’s quality of life [40].

Between 70% and 90% of patients with advanced-stage cancer experience pain. This pain can be acute or chronic, somatic or visceral, nociceptive or neuropathic, and often multiple types may coexist [40].

The type and intensity of pain vary depending on the tumor, its extent, location, the treatment received, and the patient’s pain threshold. In 80% of cases, the pain originates from the tumor itself, while in the remaining 20% it is caused by treatments [40].

According to a report by the World Health Organization (WHO) on pain and palliative care [40], it is essential to address patients’ needs throughout the entire therapeutic process, combining treatments with curative intent with supportive symptom management measures.

Thus, continuous care should begin from the moment of diagnosis and may include [40]:

**Supportive treatments**: Reduce symptoms and are administered alongside specific oncological therapies.

**Palliative care**: Aim to improve quality of life when the disease is advanced and curative treatment is no longer feasible.

**End-of-life care**: Provided when death is imminent.

### 2.1. Evaluation of Cancer Pain

A systematic assessment of pain is essential to ensure high-quality clinical care, to adjust analgesic treatment adequately, and to support research and epidemiological studies on pain [40].

Nevertheless, this assessment is often hindered because many patients do not report their pain or minimize its intensity. This attitude may stem from fear that expressing discomfort could be interpreted as a sign of therapeutic failure or disease progression. In other cases, patients avoid mentioning it so as not to distract the medical team from cancer-specific treatment [40].

Moreover, studies indicate that older patients or those with lower educational and socioeconomic levels tend to avoid reporting pain due to fear of supposedly more severe adverse effects associated with stronger analgesics [40].

It should be taken into account that pain is not only a physical sensation but also a complex emotional state, making its objective measurement difficult. Therefore, providing the patient with clear and accessible information about pain and its treatment is essential to ensure an accurate assessment. Proper evaluation of pain involves measuring intensity, understanding pathogenesis, determining its relationship with the underlying disease, assessing its impact on the patient’s quality of life, and using either a unidimensional or multidimensional approach [40].

A complete medical history should be obtained, along with a physical and psychological examination of the patient. The onset and duration of the pain should be recorded, its location and radiation determined, and any triggering or modifying factors assessed. The main objective of this evaluation is to implement a personalized treatment plan tailored to the patient’s individual needs [40].

#### 2.1.1. Dimensions of Cancer Pain

To adequately understand the pain experience in cancer patients, a comprehensive approach is essential—one that goes beyond the purely physical aspect. Pain is a complex phenomenon involving multiple interrelated dimensions, and its assessment must reflect this complexity in order to ensure effective and personalized treatment. In this regard, six key dimensions should be considered when assessing pain [40]:

**Physiological**: Includes location, duration, etiology, and type of pain (visceral, somatic, or neuropathic).

**Sensory**: Refers to pain intensity, quality, and pattern.

**Affective**: Evaluates the influence of pain on mood, well-being, anxiety, or depression.

**Cognitive**: Considers the meaning the patient attributes to pain, its relationship with the neoplasm, and coping strategies.

**Behavioral**: Assesses the impact of pain on the patient’s behavior and behavioral responses.

**Sociocultural**: Takes into account ethnic, familial, occupational, social, and spiritual aspects.

Considering these dimensions allows for a more complete and holistic evaluation, facilitating a therapeutic approach adapted to the individual needs of each patient.

It is important to highlight a key concept in pain assessment: pain should be evaluated and treated based on the patient’s description, not the healthcare team’s perception. This is essential, as pain includes subjective elements such as the patient’s impressions, perceptions, and personal experiences [40].

#### 2.1.2. Scales and Questionnaires for Cancer Pain Assessment

Scales and questionnaires are essential tools for pain evaluation. Scales offer a unidimensional perspective, focusing mainly on pain intensity, while questionnaires offer a multidimensional evaluation, including emotional, functional, and cognitive impact.

Scales are particularly useful in clinical practice due to their speed: they allow for a quick estimation of the patient’s pain intensity, often in less than 30 s. Different types of scales are used depending on the assessment goal [40].

##### Assessment of Pain Intensity

The Visual Analogue Scale (VAS) and the Numeric Rating Scale (NRS) are the most widely used tools in clinical practice. In contrast, the Verbal Rating Scale (VRS) is less commonly used, as it may be more complex for patients to interpret. Finally, in the pediatric setting, the use of the Faces Pain Scale (FPS) is common [40,41]:

**VAS:** This scale consists of a horizontal line 10 cm in length, where 0 indicates no pain, and 10 indicates very intense pain. Values from 0 to 3 are interpreted as mild pain, 4 to 6 as moderate pain, and 7 to 10 as severe pain, although some authors consider 6 already within the severe range.

**NRS:** Patients assign a value from 0 (no pain) to 10 (very intense pain), based on their perceived pain intensity.

**VRS:** Patient verbally describes their pain as none, mild, moderate, or severe. As previously mentioned, this scale is less frequently used due to its interpretive demands, especially for those patients with communication difficulties.

**FPS:** Particularly useful in pediatrics, this scale is based on the Wong–Baker Faces Pain Rating Scale, which uses facial images to avoid confusion between physical discomfort, affectivity, and emotional rejection associated with pain. The scale includes six faces, starting at point 0, which indicates no pain, and going up to 10, which represents the worst pain imaginable.

Additionally, hardware-based techniques, including quantitative sensory testing (QST) and pressure algometry, are increasingly used to detect alterations in nociceptive processing, including phenomena such as opioid-induced hyperalgesia (OIH)—a paradoxical condition in which prolonged opioid use leads to increased sensitivity to pain rather than pain relief—and individual variations in pain tolerance [42,43]. In fact, clinical evidence shows that QST—such as cold pressor and pressure pain threshold tests—can help differentiate OIH from opioid tolerance, and may signal paradoxical sensitization in patients on long-term opioids or opioid maintenance therapies [44,45]. Combining hardware-based methods with clinical tools enhances clinicians’ capacity to tailor opioid therapy more precisely, identify early signs of poor analgesic response or adverse effects, and differentiate between opioid tolerance and OIH. This multidimensional approach is particularly relevant in oncology settings where pain complexity and individual variability are significant clinical challenges.

##### Assessment of the Affective Component

One of the most commonly used tools is the affective subscale of the McGill Pain Questionnaire. In addition to this, other visual and verbal scales have been developed which, similarly to those measuring pain intensity, use lists of adjectives to describe various levels of suffering or discomfort related to pain, ranging from mild sensations to pain perceived as unbearable [40,41].

##### Assessment of Pain Location

One of the most widely used tools is the body chart, in which the patient is asked to indicate the areas where they experience pain. It has been observed that the total extent of the marked areas correlates more closely with functional limitations and reduced physical activity than with perceived pain intensity or its emotional component. This resource is easy to apply in clinical settings and is recommended as a routine tool in pain assessment [40,41].

On the other hand, questionnaires allow for a multidimensional assessment of pain, providing more precise information and increasing diagnostic sensitivity. The Brief Pain Inventory [46], the Memorial Pain Assessment Card [47], and the McGill Pain Questionnaire [48] are among the most commonly used tools.

The fastest of these tools is the Memorial Pain Assessment Card, which can be completed in under one minute. It evaluates pain intensity, pain quality, the degree of relief achieved, and the impact of pain on the patient’s mood [41].

Other questionnaires, including the Brief Pain Inventory and the McGill Pain Questionnaire, provide a broader assessment but are less practical for routine use, as they evaluate additional dimensions such as social relationships, functional activity, pain interference, modifying factors, and pain location, among others [41].

The most widely used tool in the context of CRP and neuropathic pain is the LANSS questionnaire, which is divided into two parts: a general pain questionnaire and a sensory neurological examination that allows for the assessment of allodynia, defined as the perception of pain in response to stimuli that would not normally be painful [49].

The use of scales and questionnaires provides the advantage of a more systematic and less biased evaluation, making them particularly useful in epidemiological studies, clinical trials, experimental research, and quality of care assessments. However, in everyday clinical practice, their use may be limited by time constraints and application difficulties, especially in patients with cognitive impairment or reduced functional status [40]. For this reason, most authors recommend basing pain assessment on a structured clinical interview, supplemented by a simple pain intensity scale. This combination has proven sufficient for adequate pain evaluation in routine practice, as it enables the exploration of three key aspects: pain intensity, the main characteristics of pain (such as location, temporal pattern, type, and response to previous treatments), and its impact on the patient’s quality of life [40].

### 2.2. Treatment of CRP

In 1987, the WHO established guidelines for the treatment of CRP [50]. However, according to a recent systematic review, the management of CRP remains insufficient [51,52], with approximately one-third of patients not receiving analgesia proportional to their pain intensity [52]. In this regard, a review published in 2020 reported that of the 16 million opioid prescriptions issued in the United Kingdom, only 13% were intended for the treatment of pain in cancer patients [53].

A careful assessment of pain is essential to achieve adequate control and ensure patient well-being [11]. Based on the results of this assessment, analgesic treatment is initiated with either low- or high-potency agents, following the WHO analgesic ladder [54]. In this context, a randomized multicenter study demonstrated that, in patients with moderate to severe CRP, low-dose morphine (a strong opioid) significantly reduced pain intensity compared to weaker opioids, with a rapid response and good tolerability [55]. These findings are supported by several authors who argue that many patients obtain limited benefit from weak opioids in the treatment of mild to moderate CRP. For this reason, it is recommended to initiate treatment directly with low-dose strong opioids, thereby moving immediately to the third step of the WHO analgesic ladder [56].

Regarding the route of administration, it should be chosen based on the patient’s tolerance. Nevertheless, the oral route is generally preferred, as it is the most convenient and physiological, offering a favorable risk–benefit profile and adequate bioavailability. In patients with dysphagia, or when the oral administration is not feasible, alternative routes should be considered [57].

Below are the main opioids used in the treatment of CRP, classified according to the steps of the WHO analgesic ladder.

#### 2.2.1. First-Step Analgesics

Nonsteroidal anti-inflammatory drugs (NSAIDs) or acetaminophen, with or without adjuvants, form the first step of the WHO analgesic ladder and are used to treat mild pain. There is no evidence that combining NSAIDs with second- or third-step opioids enhances analgesic efficacy [58,59,60].

Furthermore, no specific NSAID has been shown to be more effective than others in controlling pain. When selecting an NSAID, preference should be given to agents with a lower risk of gastrointestinal toxicity, convenient dosing schedules, and a favorable safety profile. In patients at high risk of gastroduodenal injury, the use of selective cyclooxygenase-2 (COX-2) inhibitors is justified [60].

#### 2.2.2. Second-Step Analgesics

Analgesics included in this group are weak or mild opioids used to manage moderate pain. Common agents in this group include codeine, dihydrocodeine, dextropropoxyphene, and tramadol [13].

Codeine is a prodrug that is metabolized in the liver into morphine. It is estimated that 5% to 10% of the administered dose is converted into morphine in most individuals [48]. This metabolic capacity varies according to individual genetic factors. Approximately 10% of the Caucasian population, 2% of the Asian population, and 1% of the Arab population are poor metabolizers, which significantly reduces the drug’s analgesic efficacy [48,56]. Conversely, ultrarapid metabolizers convert codeine to morphine at an accelerate rate, which increases the risk of toxicity even at standard doses [48,57].

#### 2.2.3. Third-Step Analgesics

This group includes drugs with greater analgesic potency, known as strong or major opioids. These include morphine, oxycodone, transdermal fentanyl, and methadone, all of which are equally effective for pain control (Table 1). The choice of agent depends on several factors, such as difficulty with oral intake ability, drug cost, or the need to optimize the balance between efficacy and adverse effects [13,60].

Strong opioids are indicated for patients with moderate to severe pain who have not achieved adequate pain relief with lower-step analgesics. Importantly, these medications do not have a therapeutic ceiling; the maximum dose is determined by individual patient tolerance [60]. Long-term use requires caution due to the risk of developing tolerance, which may necessitate increasing the dose to maintain the same analgesic effect. Additionally, abrupt discontinuation or the use of antagonists may lead to withdrawal syndrome [53].

These drugs may also offer additional benefits such as anxiety reduction, sedation, and improved rest [53]. However, all of them may are associated with potential adverse effects, including respiratory depression, nausea, vomiting, constipation, hyperalgesia, and excessive drowsiness [53].

The main opioids used for CRP included in the third step of the WHO analgesic ladder are presented below (Table 1).

##### Morphine

Morphine is an alkaloid derived from opium, first isolated in 1803 [53]. Its use in CRP management began in the 1950s, often in combination with cocaine and alcohol in a preparation known as the “Brompton cocktail” [53]. According to the WHO and the European Association for Palliative Care, morphine is the opioid of choice for the treatment of moderate to severe CRP, particularly in settings where close monitoring and titration are feasible [54]. It is widely used in both inpatient and outpatient palliative care due to its proven efficacy, affordability, and availability in multiple formulations. It is available in various administration routes: oral, parenteral, subcutaneous, intravenous, and intramuscular [61]. The wide variety of formulations and dosages provides great flexibility in managing somatic and visceral CRP [53].

Morphine is a water-soluble substance, with approximately one-third of its circulating concentration bound to plasma proteins, meaning it does not accumulate in tissues. It is also capable of crossing both the blood–brain barrier (BBB) and the placental barrier [62].

Morphine’s analgesic effect is due to its action as a potent µ-opioid receptor agonist [53]. Oral morphine has a bioavailability of approximately 30% and undergoes hepatic metabolism via UGT enzymes, producing two main metabolites: morphine-3-glucuronide (inactive) and morphine-6-glucuronide (active). The latter has a higher affinity for µ-opioid receptors and contributes substantially to analgesic efficacy. However, in patients with renal impairment, morphine-6-glucuronide may accumulate, increasing the risk of toxicity such as sedation and respiratory depression. In such cases, dose adjustments or alternative opioids like fentanyl, which is more suitable for transdermal administration and has minimal renal excretion, or oxycodone—less reliant on renal clearance—may be more appropriate. Common side effects include constipation, nausea, sedation, and pruritus [53,63].

In conclusion, despite the emergence of newer agents, morphine remains the reference opioid for cancer-related pain, against which the effectiveness and safety of other opioids are often compared.

##### Oxycodone

Oxycodone is a semi-synthetic opioid derived from thebaine and structurally similar to codeine. It is a pure agonist of the µ, δ, and κ opioid receptors, with notable activity at the κ-receptor. Clinically, oxycodone is considered a suitable second-line opioid in patients who do not respond adequately to or cannot tolerate morphine, particularly those experiencing visceral or neuropathic pain. Its high oral bioavailability and predictable pharmacokinetics make it well-suited for outpatient management, especially in palliative care settings [64].

It has an analgesic potency approximately 1.5 to 2 times greater than morphine, and is more lipophilic, though less so than fentanyl. It is available as immediate-release and extended-release oral tablets, which should not be crushed or chewed. Oxycodone has high oral bioavailability (50–80%) and is primarily metabolized in the liver by CYP3A4 (to noroxycodone) and CYP2D6 (to oxymorphone), with both active and inactive metabolites [64,65].

Common side effects include nausea, dizziness, constipation, and drowsiness. However, compared to morphine, oxycodone causes fewer hallucinations and may be better tolerated in elderly or frail patients. In addition, a fixed-dose combination of oxycodone with naloxone is also available, which reduces opioid-induced constipation without compromising analgesia, thereby improving adherence and quality of life in cancer patients [54].

Ultimately, oxycodone plays a valuable role in cancer pain management, particularly as a step-up or alternative when morphine is unsuitable.

##### Fentanyl

Fentanyl is a synthetic opioid derived from morphine that acts predominantly on μ-opioid receptors. It is approximately 100 times more potent than morphine and is characterized by its high lipophilicity, which allows for rapid penetration of the BBB, contributing to its potent analgesic effect [66].

It is mainly metabolized in the liver and intestinal mucosa by CYP3A4, producing the inactive metabolite nor fentanyl. It is considered safe in patients with renal impairment due to the absence of active metabolites [67].

Transdermal fentanyl patches are used for chronic, stable cancer pain, while intravenous or transmucosal formulations are preferred for breakthrough pain. When used transdermally, the onset of analgesic effect occurs between 12 and 18 h after application. This makes fentanyl a drug of choice for patients with stable, chronic pain who cannot tolerate oral medications or have poor gastrointestinal absorption, making it suitable for palliative care and home settings. However, it is not suitable for rapid pain relief [68]. Moreover, its slow elimination necessitates caution when switching to fast-acting analgesics, as active drug concentrations may persist for several hours. Finally, adverse effects include sedation, respiratory depression, and bradycardia [60].

In conclusion, fentanyl is not typically used as a first-line opioid but is widely employed in opioid rotation strategies or when side effects or pharmacokinetic limitations of other opioids limit their use.

##### Methadone

Methadone is a synthetic opioid with a complex mechanism of action: it is a potent μ-opioid receptor agonist, a weak NMDA receptor antagonist, and inhibits serotonin (5-HT) and norepinephrine reuptake. These properties make it particularly useful in patients with complex pain syndromes, including mixed nociceptive and neuropathic cancer pain, or in those who have developed significant opioid tolerance [69]. Methadone can be administered via oral, rectal, subcutaneous, or parenteral route [69]. It crosses the BBB easily and tends to accumulate in adipose tissue due to its high lipophilicity. It has good absorption, with an oral bioavailability ranging from 70% to 90% and a long, variable half-life (ranging from 8 to 59 h), which necessitates careful titration to avoid accumulation and delayed toxicity such as respiratory depression. It is metabolized in the liver by CYP3A4, CYP1A2, and CYP2D6 and does not produce active metabolites [70].

Overall, methadone can be a highly effective and cost-efficient option in palliative care settings, particularly in resource-limited environments or when neuropathic components are predominant.

##### Adjuvant Treatments

Some drugs, although not specifically designed for pain management, can be useful in certain clinical contexts. These medications may help reduce pain indirectly, especially when it is associated with other factors such as anxiety, sleep disorders, depression, or neurological conditions like neuralgia.

Among the main adjuvant treatments are antidepressants (both tricyclic and other types), anxiolytics, corticosteroids, antipsychotics, anticonvulsants, local anesthetics, baclofen, and bisphosphonates. These agents can enhance the effect of primary analgesics and contribute to improving the patient’s overall well-being [13].

**Table 1 ijms-26-07459-t001:** A comparative overview of commonly used strong opioids in the clinical management of CRP. The table summarizes key pharmacological and clinical characteristics, including equianalgesic oral doses, potency relative to morphine, metabolic pathways, elimination half-lives, renal safety profiles, common adverse effects, and preferred clinical use cases.

Opioid	Equianalgesic Dose (Oral)	Potency (vs. Morphine)	Metabolism	Half-Life (h)	Renal Safety	Common Side Effects	Preferred Use Cases	Ref
**Morphine**	30 mg	1×	Hepatic (UGT)	2–4	Low (accumulates)	Constipation, nausea, sedation, drowsiness, pruritus, and respiratory depression	First-line agent for CRP (somatic and visceral) Wide availability and flexibility	[53,54,63]
**Oxycodone**	20 mg	1.5–2×	Hepatic (CYP3A4/2D6)	3–4.5	Moderate	Nausea, constipation, dizziness, and drowsiness	Patients who do not respond or cannot tolerate morphine Effective in visceral/neuropathic pain; oral use	[53,54,64]
**Fentanyl**	0.1 mg (i.v.) = 30 mg morphine oral	100× (i.v.)	Hepatic (CYP3A4)	3–12 (patch: 72 h)	Safe	Sedation, bradycardia, hypoventilation, tolerance	Transdermal patches for patients with stable, chronic pain who cannot tolerate oral medications Preferred in patients with renal failure	[53,54,67,68]
**Methadone**	10 mg	Variable (0.5–1.5×)	Hepatic (CYP3A4/2D6/1A2)	8–59	Safe	Sedation, respiratory depression, unpredictable kinetics	Patients with neuropathic and nociceptive pain, or those who have developed opioid tolerance Long-acting alternative	[69,70]

Comparative characteristics of commonly used strong opioids in cancer pain management. **CRP**: cancer-related pain; **i.v.**: intravenous.

## 3. Impact of Opioid Misuse in Cancer Patients

Patients with advanced cancer are often exposed to high doses of opioids, as these are considered the first-line treatment according to the WHO analgesic ladder for CRP [50,53]. However, as treatment acceptance improves and patient survival increases, balancing adequate pain relief with the risks associated with prolonged opioid use—such as opioid use disorder (OUD)—becomes increasingly complex [17].

Patients’ beliefs and perceptions about opioids play a critical role in pain management. Some may refuse essential treatments due to fear of addiction, while others may misuse opioids and hesitate to seek help out of fear of being judged. Similarly, caregivers’ (family or friends) opinions can directly influence treatment adherence [71].

In this context, Bulls and colleagues (2023) [72] conducted a qualitative study aimed at understanding how both patients and their close contacts perceive pain, with the goal of developing strategies that improve pain control while minimizing the risk of dependence. To determine sample size, the researchers applied the criterion of thematic saturation, which involves discontinuing interviews once no new information emerges. A total of 20 patients and 11 close contacts were interviewed. Participants were adults with advanced-stage solid tumors (stage III or IV) and current or past experience with opioid treatment for CRP. Family members or friends were identified by the patients themselves and had to be available for in-depth interviews, reachable by phone, and capable of responding to questions in English [72].

It is important to note that the study had certain limitations, primarily related to the lack of racial and ethnic diversity among participants. Most were non-Hispanic White individuals from urban areas, which limits the generalizability of the findings. Additionally, many patients were recruited through recommendations from their oncologists, introducing potential selection bias. Despite these limitations, the study is pioneering in its in-depth exploration of this topic and represents a valuable contribution. Nevertheless, future research should aim to include more diverse and traditionally underrepresented populations [72].

From the interviews conducted, it was observed that many participants exhibited behaviors and attitudes intended to protect themselves from developing OUD. Several reported feeling a strong sense of responsibility in their use of medications, strictly adhering to prescribed regimens. However, it remains unclear whether such behaviors genuinely reduce the risk of OUD, and in some cases, they may even lead to inadequate pain management. Moreover, patients expressed interest in acquiring evidence-based strategies to minimize the risk of OUD during opioid treatment. Suggested strategies include specific education about prescribed medications, improved monitoring techniques, and behavioral interventions to optimize pain control. In this sense, future research should focus on developing effective interventions to reduce OUD risk and improve the management of CRP [72].

However, the study also shows that many cancer patients avoid acknowledging the risk of OUD due to fear of social judgment, which can hinder both prevention and early detection. This reluctance may be reinforced by healthcare professionals who, by downplaying the risk, fail to implement adequate monitoring strategies. In this context, it is essential to reduce stigma, improve access to care, and promote education about OUD and its prevention [72].

Additionally, several studies have demonstrated a direct association between the route of administration and the risk of opioid addiction. Intravenous (i.v.) administration is often the route of choice for patients who cannot take oral medications or when rapid pain relief is required. In hospital settings, i.v. opioids are typically administered as intermittent boluses. However, it has been observed that healthcare professionals sometimes administer these boluses faster than recommended, aiming for quicker pain relief and thereby exceeding clinical guidelines and manufacturer instructions [73].

Currently, data from two randomized clinical trials conducted in healthy, drug-naïve volunteers are available. One study found a significant association between faster administration rates and greater abuse potential, while the other did not find such a relationship [74,75]. Nonetheless, the optimal i.v. administration speed that balances effective analgesia without increasing the risk of abuse remains unclear, and no studies have yet compared abuse risk across different i.v. administration speeds.

In this context, current guidelines recommend i.v. injection over 2 min (i.e., rapid injection) [73]. However, whether this speed increases abuse potential is still uncertain. What is clear is the need to balance the analgesic benefits of rapid administration with the need to minimize addiction risk.

For this reason, Arthur and colleagues (2024) [76] recently conducted a randomized, double-blind, crossover clinical trial to evaluate the abuse potential and analgesic efficacy of i.v. hydromorphone—a semisynthetic opioid used in cancer pain management. The study compared a guideline-recommended rapid bolus injection with an experimental slow 15 min infusion (“piggyback”) in hospitalized patients at MD Anderson Cancer Center, University of Texas, who were experiencing moderate to severe pain and were not receiving chronic opioid therapy. The study authors hypothesized that slow infusion would result in lower abuse potential, fewer adverse effects, and comparable analgesic efficacy relative to the rapid injection.

No statistically significant differences in abuse potential—measured using the “drug liking” subscale of the Drug Effects Questionnaire (DEQ-5)—were found between the two administration speeds, and analgesic efficacy was also similar. This subscale is considered the most sensitive, reliable, and informative for estimating the likelihood of drug abuse, with excellent validity. Notably, the slow infusion was associated with less sedation, making it a preferred option for patients taking other sedative medications. Despite the methodological strengths of the study, limitations include a small sample size and short follow-up period [76].

In conclusion, further research is needed to determine the optimal i.v. administration speed that minimizes abuse risk without compromising pain control.

### 3.1. Assessment of the Risk of Opioid Addiction in Cancer Patients

Once the indication for opioid therapy in the management of CRP has been confirmed, it is necessary to assess the risk of misuse and potential addiction [77]. This involves reviewing the patient’s medical history, conducting a thorough clinical interview (anamnesis), and using standardized screening tools. The objective is to optimize monitoring and pain management in patients who may be at high risk of opioid misuse [77].

Below are the main standardized questionnaires used to assess the risk of problematic opioid-related behaviors, particularly in patients with chronic pain, including those with cancer.

#### 3.1.1. CAGE Questionnaire (Cut Down, Annoyed, Guilty, Eye-Opener)

This questionnaire was originally developed to detect alcohol-related problems, as alcoholism tends to be underdiagnosed without structured screening [78]. An adaptive version, the CAGE-AID (Adapted to Include Drugs), was later developed to include other substances beyond alcohol [79]. Risky consumption is considered when at least two responses are affirmative. Alcohol use is assessed because alcoholism has been found to be associated with increased risk of addiction to other substances such as tobacco, medications (e.g., benzodiazepines, opioids, stimulants), and illicit drugs [80].

#### 3.1.2. SOAPP Questionnaire (Screening and Opioid Assessment for Patients with Pain)

This questionnaire is designed to detect aberrant behaviors related to opioid use. It consists of 14 questions, each scored from 0 (never) to 4 (very often) [17]. A shorter version, the SOAPPP-SF, contains only five questions and has been used in cancer patients. In this version, a score greater than 4 is considered indicative of opioid misuse [81].

#### 3.1.3. ORT Questionnaire (Opioid Risk Tool)

The ORT is a five-item yes/no questionnaire used to predict the likelihood of opioid-related behaviors in patients undergoing opioid therapy. Patients are categorized based on the following risk factors: age (highest risk between 16 and 45 years), personal and family history of substance abuse, psychological comorbidities, and history of sexual abuse [82].

### 3.2. Factors Contributing to Opioid Misuse and Dependence

Several factors may influence the development of opioid misuse in cancer patients. These include both genetic factors—such as individual differences in opioid metabolism—and non-genetic factors, related to clinical, psychological, and environmental aspects. Analyzing these factors is essential to understanding why certain patients are more vulnerable to opioid misuse and to guiding prevention strategies and treatment personalization.

#### 3.2.1. Impact of Pharmacogenetics on Opioid Misuse and Dependence

Due to the challenge of achieving an optimal balance between analgesic efficacy and the risk of adverse effects or misuse, Bugada and collaborators (2020) [83] propose, in their review, the concept of personalized medicine. This approach could improve prescribing practices by identifying the safest and most effective dose for each patient. In doing so, optimal pain control could be achieved while minimizing both adverse effects and the risk of dependence [83].

The foundation of personalized medicine lies in selecting the most appropriate treatment for each patient, based not only on clinical history but also on genetic factors that influence pharmacokinetics and pharmacodynamics. In this context, attention has focused particularly on genetic components that affect pain perception and response to opioid therapies, with the aim of identifying the most effective medication to reduce pain and improve oncological outcomes [83].

Opioids are metabolized in the liver primarily through the cytochrome P450 (CYP450) enzyme system or via the UDP–glucuronosyltransferase (UGT) pathway. Some opioids are administered as prodrugs and must be metabolized to become pharmacologically active. Others are active upon administration and are subsequently inactivated through hepatic metabolism.

The cytochrome P450 isoenzyme 2D6 (CYP2D6) is responsible for both the activation of prodrugs and the inactivation of active opioids. CYP2D6 is involved in the metabolism of several opioids, including codeine and tramadol. The gene encoding CYP2D6 is characterized by high genetic variability, leading to variability in enzyme activity and, consequently, differences in opioid metabolism depending on the patient’s genetic profile [84]. Based on genetic variants of this gene, individuals can be classified as poor, intermediate, extensive, or ultra-rapid metabolizers. These phenotypes affect the conversion of prodrugs into their active forms, thereby influencing analgesic efficacy. Poor metabolizers may experience inadequate pain relief, potentially resulting in unjustified dose escalation. Conversely, ultra-rapid metabolizers may exhibit high plasma concentrations of the active opioids, increasing the risk of adverse effects, including euphoria-induced reinforcement that contributes to dependence [83].

Another relevant gene is catechol-O-methyltransferase (*COMT*), which influences pain perception and opioid response. The Val158Met variant of the *COMT* gene has been associated with differences in pain perception and opioid dose requirements. However, study results have been inconsistent, possibly due to small sample sizes and the complexity of genetic interactions [83].

Additionally, the *OPRM1* gene, which encodes the μ-opioid receptor, includes the A118G polymorphism that may influence opioid response. Some studies suggest that carriers of the G allele may require higher doses to achieve effective analgesia, potentially increasing the risk of addiction. Nevertheless, findings have also been variable and inconclusive [83,84].

In conclusion, genetic variability in opioid metabolism can have significant clinical implications for cancer patients, not only in terms of analgesic effectiveness but also in relation to the risk of misuse and addiction. Therefore, incorporating pharmacogenetics into clinical practice could become a key tool for identifying cancer patients at higher risk of opioid addiction and for tailoring opioid therapy in a safer and more effective manner.

#### 3.2.2. Impact of Non-Genetic Factors on Opioid Misuse and Dependence

In recent years, there has been a progressive increase in non-medical opioid use (NMOU), defined as the administration of these drugs without a medical prescription or in a manner different from what is indicated. This practice can significantly interfere with the management of CRP. In this regard, there are currently no clearly defined guidelines for identifying and appropriately managing NMOU. Although several studies have identified associated risk factors, these are not predictive in all cases: some patients with risk factors do not develop NMOU, while others without known antecedents do exhibit such behavior [85].

NMOU-related behaviors can have serious consequences for cancer patients, including increased morbidity and mortality due to the development of substance use disorders, poor cancer control as a result of non-adherence to treatment, and the presence of additional comorbidities such as mood disorders, insomnia, and opioid dependence. NMOU may manifest in various ways, including requests for early refills, sharing medication with friends or family, seeking specific drugs, obtaining prescriptions from multiple providers, or falsifying prescriptions. Given this situation, one of the main objectives of research on NMOU is to reduce the frequency of such behaviors through universal screening, minimizing opioid exposure, and conducting closer follow-up in high-risk patients [85].

In this context, Yennurajalingam and colleagues (2021) [85] aimed to identify which individual non-genetic factors are associated with NMOU in order to predict which cancer patients are at higher risk. Their approach relied on data collected during supportive care sessions conducted by physicians and nurses, using standardized questionnaires such as SOAPP-14, CAGE-AID, and ORT. After applying exclusion criteria, the analysis included 1554 patients, with a median age of 61 years (range: 52–69), of whom 816 were women (52.5%) and 1124 were White (72.3%) [85].

This was the first study to assess the frequency of NMOU-related behaviors in cancer patients. Results showed that approximately 19% of patients receiving long-term opioid therapy for cancer pain exhibited NMOU behaviors, with early refill being the most common. Furthermore, the study identified that patients who were single or divorced, had high SOAPP scores, experienced severe pain, and were receiving higher daily opioid doses were at greater risk of engaging in NMOU. These factors may be useful for establishing clinical risk profiles [85].

In addition, several other non-genetic risk factors have been associated with opioid addiction, including a positive CAGE questionnaire result, a personal and/or family history of drug use, alcoholism, tobacco use, younger age, worse scores on the ESAS scale (indicating higher symptom burden), better functional status, higher daily morphine equivalent doses, psychiatric or psychological comorbidities such as depression and anxiety, and a history of sexual abuse [85].

## 4. Treatment of Opioid Use Disorder

Although the opioid overdose crisis is well documented, treatment for opioid addiction remains less frequently addressed. This lack of information often results in many patients receiving care only in emergency departments.

According to the U.S. Substance Abuse and Mental Health Services Administration (SAMHSA) [86], the majority of patients with OUD do not access treatment and recovery programs, primarily due to barriers such as cost and difficulty finding appropriate treatment centers and programs. In light of this, it is a public health priority to promote the use of medications that reverse overdoses and to ensure equitable access to withdrawal and recovery treatments.

Historically, the response to the opioid crisis focused more on public safety than public health. However, beginning in 2012, the focus began to shift toward research and the development of treatments, rather than solely on identifying causes, routes of use, or criminalizing addiction.

In oncology, as CRP decreases, opioid dosages are gradually reduced. However, this tapering process may lead to withdrawal syndrome or the emergence of OUD—an increasingly recognized issue. In fact, between 2002 and 2014, admissions to specialized opioid abuse treatment centers quadrupled. Despite this trend, only one in six patients with OUD received specialized care.

### Buprenorphine-Naloxone (Suboxone)

In 2002, in response to the opioid crisis, the U.S. Department of Health approved medication-assisted treatment (MAT), which involves substituting the addictive opioid with regulated medications. One of the most widely used options is the combination of buprenorphine (a partial μ-opioid receptor agonist) and naloxone (an opioid receptor antagonist, particularly at μ-receptors), marketed as Suboxone. This combination is considered a safe and effective treatment for detoxification and long-term OUD management. In Europe, this pharmacological combination was approved by the European Medicines Agency (EMA) in 2006 [87].

The buprenorphine-naloxone combination has demonstrated efficacy comparable to methadone in managing withdrawal symptoms and offers the advantage of being prescribable by non-specialist physicians. This broadens access to treatment for OUD patients who may be unwilling or unable to attend specialized addiction programs [87].

The efficacy of buprenorphine-naloxone was evaluated in a Cochrane review published in 2003 [87]. The treatment followed the MAT model, combining Suboxone administration with psychosocial interventions for the management of OUD. Three key indicators were used to assess its effectiveness: treatment adherence (in years), dosage administered, and the ability to achieve complete abstinence or reduce the use of opioid agonists following CRP therapy. The review also evaluated pain intensity before and during buprenorphine-naloxone treatment using validated scales (CMSAS and Brief Pain Inventory). The sample included cancer patients with a mean age of 55, comprising both chronic pain survivors and patients with active metastatic disease. Results showed that the buprenorphine-naloxone combination was effective in both controlling CRP and reducing opioid craving and misuse. Moreover, higher doses administered over longer periods were associated with improved adherence and reduced relapse rates. For these reasons, this combination is proposed as a valid therapeutic option for patients with OUD during or after treatment for CRP [87].

## 5. Clinical Considerations and Perspectives on Opioid Use in Oncology

In the oncology setting, there is a high prevalence of opioid use for CRP management, which presents a major challenge given the risk of misuse and the potential development of OUD, particularly in cases involving long-term treatment, rapid administration routes, lack of effective analgesic alternatives, and, most importantly, individual genetic vulnerability. In this context, pharmacogenetics may support safer and more personalized prescribing by improving analgesic efficacy, reducing adverse effects, and minimizing the risk of dependence. Moreover, the buprenorphine-naloxone combination has proven to be an effective therapeutic option for patients with OUD, providing adequate pain control with lower abuse potential.

Of note, a direct comparison of clinically used opioids for CRP reveals that no single agent is universally superior. Morphine remains the gold standard due to its proven efficacy and broad accessibility, but its active metabolites can accumulate in patients with renal impairment. In such cases, fentanyl—due to its high lipophilicity and lack of active metabolites—is preferred. Oxycodone, with significant activity at the κ-opioid receptor, has shown superior efficacy in certain cases of visceral or neuropathic pain. Additionally, the fixed combination of oxycodone with naloxone may help reduce opioid-induced constipation. Methadone, while effective in both nociceptive and neuropathic pain, requires careful monitoring due to its long and variable half-life. Therefore, the optimal opioid must be selected based on individual clinical factors, including pain type, organ function, pharmacokinetics, and potential for misuse.

In this context, proper assessment of CRP is essential to delivering personalized and effective treatment. However, this is often hindered by patients’ reluctance to report their pain, whether due to fear of burdening the medical team or concerns about receiving stronger, potentially more addictive treatments. Therefore, it is vital to educate patients from the time of diagnosis about the importance of pain communication and to destigmatize opioid use. When used under medical supervision, opioids do not necessarily pose a real risk of addiction and can significantly improve quality of life.

The therapeutic plan should be based on the patient’s perceived pain rather than on standardized protocols due to the subjective nature of CRP. In this regard, the systematic use of validated scales and questionnaires can enhance pain assessment and management. Despite being underuse in clinical practice due to time constraints, several studies have shown that these tools contribute to optimizing treatment and improving patient-centered care.

Beyond the risk of addiction, hospital opioid use has also been associated with relevant adverse effects, such as respiratory depression, hypotension, nausea, vomiting, longer hospital stays, and increased risk of readmission. In response, therapeutic guidelines have been established to restrict opioid use to clinically justified situations. However, these guidelines are not always easily accessible or widely disseminated among healthcare professionals, which hinders their systematic implementation. Therefore, it is necessary to promote their integration into routine clinical practice, alongside healthcare provider training and access to updated clinical protocols. This would support safer and more ethical decision-making that balances the duty to relieve pain with the responsibility to prevent misuse and dependence.

Although statistically significant results have not yet been established, some studies suggest a potential relationship between i.v. administration speed and the risk of abuse. The hypothesis is that slower administration may reduce this risk. What has been demonstrated is that slower infusion results in less sedation, while faster administration does not significantly alter analgesic efficacy, as both speeds achieve similar pain control. Further research with larger cohorts and longer follow-up periods is needed to clarify the influence of administration route and speed on the addictive potential of opioids.

In addition to administration parameters, individual genetic variability must be considered. Pharmacogenetics emerges as a key tool for treatment personalization, particularly in oncology. The *CYP2D6* gene polymorphism affects opioid metabolism and may increase the risk of problematic use. Although studies on the *COMT* and *OPRM1* genes suggest possible associations with pain sensitivity and opioid response, larger and more robust investigations are needed to yield conclusive evidence. These factors reinforce the need for an individualized approach that considers not only the clinical characteristics of pain but also each patient’s genetic profile.

Incorporating pharmacogenetic models into clinical practice could enhance the personalization of analgesic treatments, allowing clinicians to anticipate both therapeutic response and the risk of toxicity or addiction. Combined with active monitoring strategies—including early screening, identification of risky behaviors, and targeted clinician training—this approach is essential to minimize the risks associated with prolonged opioid use. Current evidence points toward a paradigm shift in CRP management, based on a multidimensional framework that integrates surveillance, education, and personalized medicine.

The findings reviewed underscore the importance of balancing effective CRP control with addiction risk prevention. In this context, the buprenorphine-naloxone combination stands out as a promising therapeutic option for patients with a history of OUD, offering effective pain control while reducing relapse and withdrawal risks. Additionally, because it can be prescribed by non-specialist physicians, this option facilitates broader integration in oncology care, supporting more accessible and inclusive pain management strategies. When combined with psychosocial interventions, it may represent a key component in the management of OUD among cancer patients.

In cases where NMOU behaviors are observed—such as early refill requests—there is a clear need to implement early detection and control strategies. Measures such as universal screening, strict supervision and limitation of opioid supply, and close clinical monitoring through safe prescribing protocols could significantly reduce the incidence of such behaviors. At the same time, psychosocial support for both patients and families is critical to encourage responsible and safe medication use. Nonetheless, further oncology-specific research is needed to deepen understanding of this issue and to develop effective prevention and intervention protocols.

Looking ahead, it is essential to promote new studies focused on oncology patients at risk of developing OUD, as current evidence is limited and insufficiently representative. Many existing studies lack racial and ethnic diversity in their samples. Additionally, further research is needed into alternative or complementary therapies to opioids, which may offer effective pain control with a lower risk of addiction.

Finally, it is important to highlight that despite the availability of international guidelines and evidence-based strategies for opioid prescribing in cancer patients, multiple barriers continue to hinder their implementation across healthcare systems. These include disparities in healthcare infrastructure, limited availability of pain specialists, lack of access to essential medications, and regulatory restrictions that vary by region. In low- and middle-income countries, opioids are often unavailable or underprescribed due to logistical constraints, fear of diversion, and inadequate training among healthcare providers [88,89,90]. Even in high-income countries, barriers persist. These include insufficient clinician education in pain management, stigma associated with opioid use, and lack of integration of pain management protocols into routine oncological care [88,89,90].

The integration of pharmacogenetic tools, while promising, is also limited by cost, availability of testing infrastructure, and lack of standardized interpretation across institutions. Addressing these barriers requires coordinated action, including provider training, patient education, regulatory reform, and investment in health system resources [91,92]. In conclusion, a global perspective is essential to ensure that advances in pain management are translated into equitable access and quality care for cancer patients across diverse healthcare settings.

## 6. Conclusions

In conclusion, this review highlights that opioids remain essential for the treatment of CRP, particularly in advanced stages of the disease. Their efficacy in alleviating pain is well established; however, their use is not without clinical, social, and ethical risks. In the absence of rigorous protocols, specialized training, and appropriate follow-up, opioid therapy can lead to serious adverse effects and increased vulnerability to developing OUD.

This review also identified and described the main opioids used in the oncology setting—such as morphine, oxycodone, fentanyl, and methadone—and their indications according to pain intensity and the patient’s clinical needs. Despite their utility, factors such as route and speed of administration, treatment duration, history of prior drug use, individual treatment response, and, most importantly, genetic variability must be considered, as they may influence both therapeutic efficacy and the risk of toxicity or dependence.

Additionally, existing strategies for monitoring and controlling opioid use were explored, including the use of pain assessment tools, early detection of high-risk behaviors (NMOU), and the integration of pharmacogenetic models. The importance of proper healthcare professional training, psychosocial support, and patient education was emphasized to promote more responsible and safer opioid use. In this regard, the buprenorphine-naloxone combination has proven to be a promising, safe, and effective therapeutic alternative, providing adequate pain relief with reduced abuse potential and relapse risk.

Finally, it is concluded that further research should focus on oncology patients at risk of opioid misuse, while promoting the development of safe prescribing protocols and exploring therapeutic alternatives with lower addictive potential. Only through an integrated, multidisciplinary, and personalized approach can opioid use in oncology be optimized, ensuring both adequate pain control and the prevention of associated risks.

## Figures and Tables

**Figure 1 ijms-26-07459-f001:**
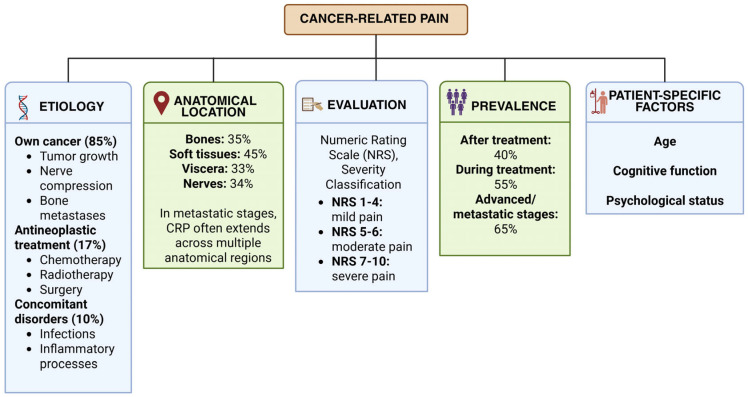
**Key factors of cancer-related pain**. The figure summarizes the main aspects involved in the assessment and understanding of cancer-related pain (CRP), including etiology, anatomical location, evaluation through the Numeric Rating Scale (NRS), prevalence across different stages of treatment, and patient-specific factors. This multidimensional approach supports a comprehensive and personalized pain management strategy in oncology. Created in BioRender. Ruiz de Porras, V. (2025). https://BioRender.com/1aov67t.

**Figure 2 ijms-26-07459-f002:**
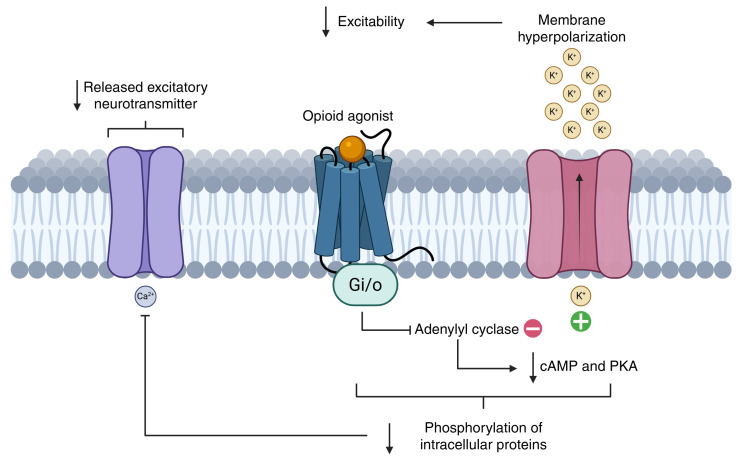
**Cellular mechanism of action of opioid receptors.** Binding of an opioid agonist to the µ-opioid receptor activates Gi/o protein signaling, leading to inhibition of adenylyl cyclase and a subsequent decrease in intracellular cyclic adenosine monophosphate (cAMP) levels and protein kinase A (PKA) activity. This results in reduced phosphorylation of intracellular proteins. Concurrently, opioid receptor activation inhibits voltage-gated calcium channels (reducing excitatory neurotransmitter release) and opens potassium channels, causing membrane hyperpolarization. These combined effects lead to reduced neuronal excitability and diminished synaptic transmission. +: activation; −: inhibition. Created in BioRender. Ruiz de Porras, V. (2025). https://BioRender.com/bx6zmb5.

**Figure 3 ijms-26-07459-f003:**
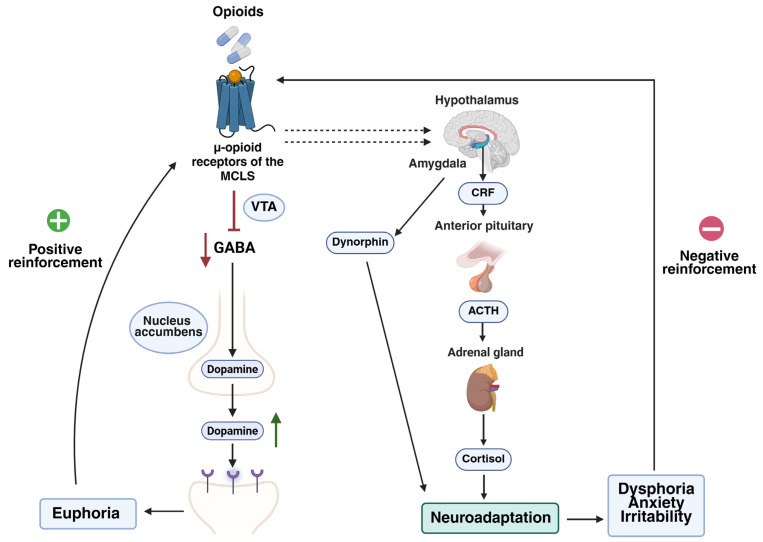
**Neurobiological mechanisms of opioid addiction.** Opioids bind to µ-opioid receptors in the mesocorticolimbic system (MCLS), leading to inhibition (**Ͱ**) of GABAergic neurons in the ventral tegmental area (VTA). This disinhibits dopaminergic neurons, causing increased (**↑**) dopamine release in the nucleus accumbens, which promotes euphoria and positive reinforcement. With chronic opioid use (--->), neuroadaptations occur via activation of the amygdala and the hypothalamic–pituitary–adrenal (HPA) axis, resulting in the release of corticotropin-releasing factor (CRF), adrenocorticotropic hormone (ACTH), and cortisol. This stress response, along with the release of dynorphin, contributes to negative emotional states such as dysphoria, anxiety, and irritability, reinforcing drug-seeking behavior through negative reinforcement. Created in BioRender. Ruiz de Porras, V. (2025). https://BioRender.com/9vft0z4.

## Data Availability

Not applicable.

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
