# Peer review of "Opioid Use in Cancer Pain Management: Navigating the Line Between Relief and Addiction"

_ijms, 2025, doi:10.3390/ijms26157459_

Round 1
Reviewer 1 Report
Comments and Suggestions for Authors
Most of the introduction part is talking about pain in general including cancer. It would be better to focus on the title of the review “Opioid Use in Cancer Pain Management”.
The review should focus on the role of different opioid drugs in management of cancer related pain clinically.
The review is missing comparison of the clinically used doses of different opioid drugs and their reported side effects besides which opioid drug is the best with less side effects.
correct 2.2.3.1. Morfine to Morphine
Comments on the Quality of English Language
The English language should be revised carefully for typos and corrected.
Reviewer 2 Report
Comments and Suggestions for Authors
Hello, dear authors!
The review is dedicated to the current topic - the use of opioids to treat pain in cancer patients. I am impressed by an integrated approach to the problem where the authors discuss not only clinical pharmacology, but also raise issues of rational pharmacotherapy.
Nevertheless, I would like the authors to pay attention to the methods of evaluating pain. It can be psychological express tests, as well as hardware techniques. This inclusion is important, as it allows you to discuss tolerance to pain relief on the one hand, and on the other, to emphasize the problems of hyperalgesia and personal tolerance.
It will also be useful to make a small section about potential barriers to introduce recommended strategies in different healthcare systems
In this regard, it will be appropriate to make a graphic abstract, as well as supplement the list of literature
Round 2
Reviewer 1 Report
Comments and Suggestions for Authors
The authors responded adequately to the comments and the manuscript is now suitable for publication in the current revised form.
Thank you